# How Czech Adolescents Perceive Active Commuting to School: A Cross-Sectional Study

**DOI:** 10.3390/ijerph17155562

**Published:** 2020-08-01

**Authors:** Michal Vorlíček, Petr Baďura, Josef Mitáš, Peter Kolarčik, Lukáš Rubín, Jana Vašíčková, Ferdinand Salonna

**Affiliations:** 1Institute of Active Lifestyle, Faculty of Physical Culture, Palacký University, 771 11 Olomouc, Czech Republic; petr.badura@upol.cz (P.B.); josef.mitas@upol.cz (J.M.); peter.kolarcik@upol.cz (P.K.); lukas.rubin@upol.cz (L.R.); jana.vasickova@upol.cz (J.V.); ferdinand.salonna@upol.cz (F.S.); 2Department of Health Psychology and Research Methodology, Faculty of Medicine, P.J. Safarik University, Trieda SNP 1, 040 11 Košice, Slovakia; 3Department of Physical Education and Sport, Faculty of Science, Humanities and Education, Technical University of Liberec, 461 17 Liberec, Czech Republic

**Keywords:** school-aged children, cycling, walking, active transport, misperceptions, social norms

## Abstract

To achieve a healthy lifestyle, adolescents must be physically active and meet physical activity (PA) guidelines. One of the most natural ways of increasing the amount of PA is active commuting (AC) to school. Recent reviews suggest that peer norms have the potential to shape PA during adolescence in particular. Thus, our primary aim was to investigate whether Czech adolescents misperceive their peers’ AC behaviors and attitudes towards AC. Our dataset comprised cross-sectional data on 1586 adolescents aged 11–15 years. Basic descriptive statistics, chi-square tests, and correlation analyses were used to analyze the data. Regarding traveling to school, 68% of the Czech adolescents in this study are daily active commuters (walking, cycling, or riding a scooter or skateboard). Less than half of the respondents believed that most of their classmates were commuting to school actively almost daily. The students who believed that most of their classmates commuted to school actively had significantly higher chances of being regular active commuters themselves. The results showed that most of the Czech adolescents misperceived the AC norms of their peers. Thus, there could be potential in using a social norms approach aimed at increasing the level of AC in Czech adolescents through targeted interventions.

## 1. Introduction

Lack of daily physical activity (PA) has been proven to have significant negative effects on human health [1,2,3]. Higher volumes of PA correlate positively with a decreased likelihood of numerous cardiovascular and lifestyle diseases [4,5], as well as cancer [6]. Already in childhood and adolescence, PA plays an important role in the prevention and treatment of childhood obesity, which, especially at the adolescent stage of development, influences behavioral habits during adulthood [7,8]. To achieve the health benefits of PA, children and adolescents should obtain at least 60 min of moderate-to-vigorous PA daily [9,10,11]. One of the ways to contribute to obtaining this amount of PA is through active commuting (AC) to/from school [12,13,14,15,16,17]. Previous findings have confirmed that AC to school contributed to the successful achievement of health recommendations for PA in Czech adolescents [18].

AC is usually defined as walking, cycling, and various other forms of transportation (e.g., in-line skating, skateboarding, or cross-country skiing), which uses human energy for transportation to and from school or work [14,19]. It represents a simple method of incorporating PA in the form of walking or cycling into everyday life. Despite its advantages, AC is still underused, which might be due to less supportive built environments [20], parents’ preferred travel options [12,13], cultural differences [21], and/or other factors [22]. The proportion of Czech adolescents commuting to school actively has been decreasing. According to the findings from an HBSC study (Health Behaviour in School-aged Children), the proportion of children AC (walking and cycling) to school decreased significantly, from 74.3% in 2006 to 53.4% in 2014 [23]. Another study [24] found an even higher decrease (47%) in AC between the years 2001 and 2011, from an absolute rate of 49.1% to a decreased rate of 26%. To promote AC, it is essential to understand which factors influence the choice of AC [25].

Previous research on transportation preferences focused primarily on how individuals weigh the expense and advantages of a mode of transport. The factors involved in consideration primarily related to time, money, and convenience [26]. Nevertheless, personal transportation choices are not fully explainable using rational choice variables [26,27]. To explain transportation behavior, the social context and subjective experiences also have to be examined [27,28,29].

Human behavior is profoundly influenced by social norms across the lifespan [30]. Several theories of health behavior imply that social comparisons and perceived social norms affect behavior in a wide variety of domains [31,32,33] According to the meta-analysis of Sheeran et al. [34], interventions that modify attitudes, norms, and self-efficacy are effective in promoting changes in health behavior. They found that experimentally induced changes in attitudes, norms, and self-efficacy all led to medium-sized changes in intention (d+ = 0.48, 0.49, and 0.51, respectively), and engendered small to medium-sized changes in behavior (attitudes-d+ = 0.38, norms-d+ = 0.36, self-efficacy-d+ = 0.47).

Recent reviews suggest that peer norms have the potential to shape PA during adolescence in particular [35,36,37]. The effect of social norms on adolescent health-related behaviors was best described in risky behaviors, such as alcohol consumption, tobacco use, and substance use [38,39,40,41]. Nevertheless, some previous studies indicate that approaches based on social norms could be a suitable method of intervention to combat physical inactivity or promote PA [37,42,43,44].

The Social Norms Approach (SNA) is the most widely used intervention strategy for promoting positive health-related behaviors on the basis of the effect of social norms [45]. The approach is founded on the premise that individuals misperceive their peers’ behaviors and attitudes, with evidence of under- and over-estimations of behaviors and peer approval of a range of positive and negative behaviors, respectively [46]. The greater these misperceptions are, the more likely an individual is to engage in negative behaviors, such as consuming larger amounts of alcohol; and reduce positive behaviors, such as eating healthily, using sun protection, or being physically active.

Given that the discrepancy between actual and perceived levels of behavior is crucial to the application of the SNA, our primary aim was to investigate whether Czech adolescents misperceive their peers’ AC behaviors and attitudes towards AC. Furthermore, we aimed to investigate the associations between their perception of AC and their actual AC behaviors and whether these associations differed with gender and class grade.

## 2. Materials and Methods

### 2.1. Procedure Sample and Participant Selection

The present cross-sectional study involved 1586 adolescents (52% of them boys) aged 11–15 years (mean 12.95 ± 3.6 years) from 12 randomly selected schools (Grades 6–9) from the Czech Republic. Schools with a specific focus on sport and schools for pupils with special educational needs were not recruited.

The study was approved under reference number 38/17 by the Ethics Committee of the Faculty of Physical Culture, Palacký University Olomouc, which is governed by the ethical standards set out in the World Medical Association Declaration of Helsinki and its later amendments. All the participants’ parents were informed about the research in advance, and permission was granted in the form of signed informed consent. Measurement was voluntary, and no incentives were provided in return for participation. Less than five percent of the adolescents opted out of the data collection.

### 2.2. Assessments and Measures

The questionnaire on activities was developed for the purpose of the SONIAA (Social norms intervention in the prevention of excessive sitting and the promotion of physical activity among Czech adolescents) project and was compiled from various internationally recognized surveys (e.g., HBSC [47], IPEN Adolescent (International Physical Activity and the Environment Network) [48,49], and YAP (Youth Activity Profile) [50,51]). The questionnaire was designed to measure various domains of PA and sedentary behavior, with a focus on the respondents’ own behaviors and attitudes as well as their perception of PA and sedentary behavior in their peers. For the purposes of this study, we used a question on AC to school over the past seven days which was adapted from the YAP (Youth Activity Profile) as a feasible and valid survey instrument at the group level [52]: “How many days did you walk, cycle, or ride a scooter/skateboard to school?” The students could select a response from five options, ranging from “zero days (never)” to “four or five days (almost every day)”.

The perceptions of peers’ AC were assessed using the following item: “How many days do you think most of your classmates walked, cycled, or rode a scooter/skateboard to school?” Moreover, we collected data on perceived peer attitudes towards peer AC using the question, “Do you think it is fine like that? We are asking you about your opinion on active time spent on the way to school by the majority of your classmates.” The response options were: “Yes, it is fine”, “Rather fine”, “I don’t know”, “Rather not fine”, and “No, it is not fine”. The results concerning the perceived peer attitudes are not presented in this paper.

### 2.3. Procedure

The data were collected from 2017 to 2018 in regular school weeks during the spring and autumn seasons. The pupils filled in an electronic questionnaire at school during class under the supervision of teachers and researchers.

### 2.4. Data Processing

Statistical data processing was performed using the IBM SPSS Statistics software, version 23 (IBM, Armonk, NY, USA). First, we computed descriptive statistics for the respondents’ rates of self-reported AC (conducted approximately once a day) and their perception of their peers’ AC in the total sample. Subsequently, the same method was used after stratification by gender and grade. Statistical differences in the respondents’ own frequency of AC (independent variable) and the perceived frequency of their peers’ AC (dependent variable) by gender and grade were assessed by Chi-squared tests and the independent samples Mann–Whitney U-test. Pearson’s correlation coefficient was used to test the relationship between the respondents’ actual AC and the perceived level of their peers’ AC.

## 3. Results

When traveling to school, 68% of the Czech adolescents reported AC almost daily (four to five days a week). Nearly 20% of the adolescents indicated that they never commuted to school actively (Table 1). AC to school was more common among the girls (72%) than the boys (64%). According to the independent samples Mann–Whitney U Test, this is a statistically significant difference (*p* < 0.001). We did not observe significant differences across grades in the boys (χ^2^ = 10.61; *p* = 0.563) and girls (χ^2^ = 16.38; *p* = 0.174).

Regarding the perceived AC levels in Czech adolescents, 10% of the respondents believed that the majority of their classmates never commuted to school actively. Less than half of the respondents (43%) believed that most of their classmates commuted to school actively four or five days a week. However, these figures are gender- and grade-dependent (Table 2). While 49% of the girls believed that their peers commuted to school actively, this opinion was shared by only 39% of the boys (*p* < 0.001). Thus, the boys underestimated their classmates’ AC slightly more than the girls. There were significant differences across grades in the girls (χ^2^ = 23.99; *p* = 0.021), but not in the boys (χ^2^ = 6.04; *p* = 0.914).

The average difference between the students’ perceived and actual norm of AC is 25% (64% vs. 39%) for the boys (χ^2^ = 103.28; *p* < 0.001) and 23% (72% vs. 49%) for the girls (χ^2^ = 84.06; *p* < 0.001). There is a smaller difference across younger (sixth- and seventh-grade) students (62% vs. 43%; χ^2^ = 60.69; *p* < 0.001) than older (eighth- and ninth-grade) students (70% vs. 44%; χ^2^ = 102.94; *p* < 0.001). The observation of the data by grade showed that 42% of the sixth-graders, 44% of the seventh-graders, 42% of the eighth-graders, and 46% of the ninth-graders believed that the majority of their classmates used AC nearly daily. In all grades, this was a lower percentage than the real (self-reported) situation (Figure 1).

The students who believed that most of their classmates commuted to school actively were also significantly more likely to be regular users of active forms of transportation for their commuting to school. This proved true in both the boys (χ^2^ = 29.87; *p* < 0.001) and girls (χ^2^ = 12.08; *p* = 0.001) (Figure 2). Correlational analysis underlined these results. There was a positive correlation between the actual AC and perceived AC (r = 0.265; *p* < 0.001).

## 4. Discussion

Our primary aim was to investigate whether Czech adolescents misperceived their peers’ AC behaviors and attitudes towards AC. We were interested in this in order to assess whether the Social Norms Approach (SNA) would be an appropriate means for intervention to promote AC among adolescents. According to Perkins [53], two conditions must first be satisfied to apply the SNA appropriately:

A.There must be misperceptions regarding actual behavior and perceived behavior—this means that there must be a difference between what people do and what they think other people do or believe. This discrepancy in perception has to be in the direction of the overestimation of problem behavior (in our case, passive transport). If there is no difference, the SNA is not appropriate.B.At least half of the population must behave “correctly”—because a social norms approach assumes that individuals want to be normal, if the majority behave in a way that is harmful, a social norms message campaign might encourage the harmful behavior. Thus, if over a half of the population behaves in a way that is contrary to the intervention, the SNA is not appropriate.

Our results indicate that 68% of the Czech students in our study commute to school actively on a daily basis. This means that it is the objective norm among Czech adolescents. Nevertheless, the majority of the students in our survey, 56%, do not consider their classmates daily active commuters; thus, the perceived norm among Czech adolescents is not to be a daily active commuter. These results give schools the opportunity to educate students regarding the reality of AC amongst their peers. This might give adolescents the impression that AC is common because most of their classmates are using this form of commuting. This approach could be especially successful in adolescent boys. The boys considerably underestimated their peers’ AC usage. Only 39% of the boys were able accurately to assess the situation, which is that most of their classmates commute to school actively four to five days a week. As 68% of the students commute actively on a regular basis, 61% of the boys incorrectly underestimated the AC levels of their classmates. In girls, 49% assessed the situation about AC in their peers correctly, and thus 51% incorrectly underestimated the AC levels of their classmates. The difference is bigger for the boys. This could be explained by a review study in adolescent girls [54] showing that young girls’ PA and nutrition are affected by gender norms, social influence, institutions, and environments.

AC behavior in these Czech adolescents is comparable to that of Swedish participants in the study by Chillón et al. [55], where 67% of the Swedish adolescents commuted to school actively. On the other hand, Estonian adolescents participating in the same study commuted less (56%). In the ENDORSE (Environmental Determinants of Obesity in Rotterdam SchoolchildrEn) study [56], almost 50% of the Dutch adolescents reported commuting to school on most school days by riding a bike or walking. Low rates of AC to school were also reported in Australia, where less than 46% of the children walked or cycled to school [57]. In our study, more girls and older ones commuted to school actively on most school days compared to Czech boys and younger adolescents. Similar results for Czech boys and girls were found in a study by Pavelka et al. [58], where 60–69% of the schoolchildren participating in the HBSC study commuted to school actively.

Our findings may indicate that the principle of “judging others by one’s own standards” is present among Czech adolescents. Those who perceive others as inactive commuters are less likely to commute actively themselves and vice versa. In line with social cognitive theory [59], individuals may change their behavior on the basis of their perception of their peers’ behavior. “Correcting this misperception” thus appears to provide an opportunity to use this discrepancy to promote PA. Feedback based on descriptive norms has the potential to facilitate an increase in the numbers of AC adolescents. This theory is supported by the results of a recently conducted study involving a randomized-controlled trial to increase the daily step count of young adults, where intervention aimed at descriptive norm feedback was successful [60].

This study has several strengths and limitations. One limitation is that it is based on self-reported data from adolescents, and so the reported prevalence may be biased. The participants consecutively reported their own AC performance and their perception of their peers’ AC levels. Exclusively using self-reported data leaves room for many alternative possibilities that were not recorded (e.g., social desirability, the false consensus effect, accessibility, consistency, etc.). The study does not adjust its analysis for the travel distance from home to school. Neither does it deal with AC from school. Moreover, we are not able to analyze the direction of the relationship between the respondents’ own actual level of AC and the perceived AC because of the cross-sectional nature of our data. However, to the best of our knowledge, this is one of few studies focusing on the AC of adolescents to school in the context of social norms. It includes a sample of sixth- to ninth-grade pupils, and is the first study in the Czech Republic to do so.

Our results show that a discrepancy exists between the perceived and actual social norms of AC among Czech adolescents. Misperception is greater among boys than girls. Thus, we can conclude that there is potential to apply targeted interventions using the social norms approach aimed at increasing or reaffirming the actual levels of AC of Czech adolescents. Future studies will require controlled trials to perform an adequate examination of the effects of associated travel behavior, as well as longitudinal designs aimed at examining changes in the sociodemographic and environmental infrastructure and their relations with long-term trends in AC patterns in children. The transition from childhood to adolescence also offers an exciting opportunity to explore how these changes could affect travel behavior.

## 5. Conclusions

Our primary aim was to investigate whether Czech adolescents misperceived their peers’ AC behaviors and attitudes towards AC to assess the potential of interventions based on the SNA. Our study points out that there is a discrepancy between the actual levels of AC and the perceived levels of peers’ AC. Adolescents overestimate the prevalence of passive transport, despite the fact that most of them commute actively. These findings indicate that there might be room for targeted intervention based on the SNA to increase the PA of adolescents or at least strengthen their actual positive behavior.

## Figures and Tables

**Figure 1 ijerph-17-05562-f001:**
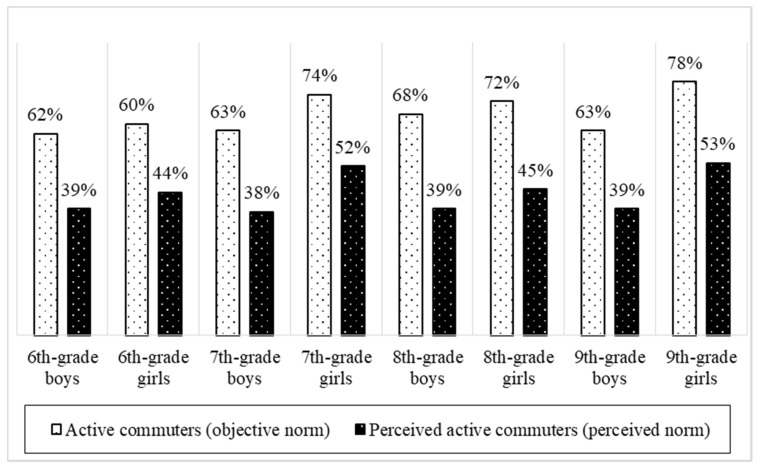
Misperceptions of the level of students commuting to school actively among Czech adolescents, according to gender and school grade.

**Figure 2 ijerph-17-05562-f002:**
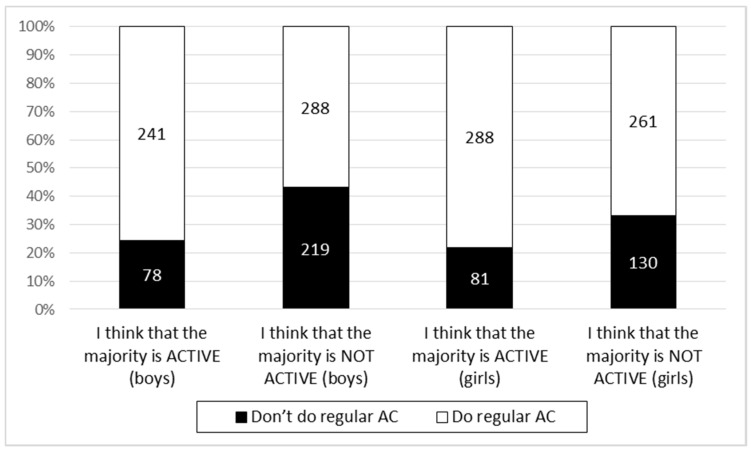
Absolute numbers of adolescents with/without regular AC to school by different perceived norms.

**Table 1 ijerph-17-05562-t001:** Actual active commuting to school among Czech adolescents according to gender and class grade.

		0 Days		1 Day		2 Days		3 Days		4 or 5 Days		χ^2^ (*p*)
	Grade	n	%	n	%	n	%	n	%	N	%	
total		308	19.4	77	4.9	50	3.2	73	4.6	1078	68.0	
boys		186	22.5	48	5.8	24	2.9	39	4.7	529	64.0	16.06 (0.002)
girls		122	16.1	29	3.8	26	3.4	34	4.5	549	72.2	
boys	6th	58	26.2	12	5.4	7	3.2	6	2.7	138	62.4	10.61 (0.563)
	7th	44	20.5	18	8.4	5	2.3	13	6.0	135	62.8	
	8th	42	21.0	7	3.5	5	2.5	10	5.0	136	68.0	
	9th	42	22.1	11	5.8	7	3.7	10	5.3	120	63.2	
girls	6th	40	19.0	14	6.7	6	2.9	13	6.2	137	65.2	16.38 (0.174)
	7th	32	16.6	6	3.1	6	3.1	6	3.1	143	74.1	
	8th	29	16.9	5	2.9	8	4.7	6	3.5	124	72.1	
	9th	21	11.4	4	2.2	6	3.2	9	4.9	145	78.4	

**Table 2 ijerph-17-05562-t002:** Perceived active commuting to school among Czech adolescents according to gender and class grade.

		0 Days		1 Day		2 Days		3 Days		4 or 5 Days		χ^2^ (*p*)
	Grade	n	%	n	%	n	%	n	%	n	%	
total		158	10	110	6.9	240	15.1	390	24.6	688	43.4	
boys		117	14.2	71	8.6	134	16.2	185	22.4	319	38.6	51.13 (<0.001)
girls		41	5.4	39	5.1	106	13.9	205	27.0	369	48.6	
boys	6th	29	13.1	18	8.1	41	18.6	47	21.3	86	38.9	6.04 (0.914)
	7th	31	14.4	15	7.0	33	15.3	55	25.6	81	37.7	
	8th	30	15.0	18	9.0	27	13.5	47	23.5	78	39.0	
	9th	27	14.2	20	10.5	33	17.4	36	18.9	74	38.9	
girls	6th	23	11.0	14	6.7	29	13.8	51	24.3	93	44.3	23.99 (0.021)
	7th	7	3.6	8	4.1	28	14.5	50	25.9	100	51.8	
	8th	7	4.1	10	5.8	27	15.7	50	29.1	78	45.3	
	9th	4	2.2	7	3.8	22	11.9	54	29.2	98	53.0	

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
