# Peer review of "How Czech Adolescents Perceive Active Commuting to School: A Cross-Sectional Study"

_ijerph, 2020, doi:10.3390/ijerph17155562_

Round 1

Reviewer 1 Report

Title: “The Role of Perceived Norms in Adolescents Actively Commuting to Schools” What is your outcome? It’s not clear at first glance from reading your title.

Abstract: Rephrase the first statement to: A lack of daily physical activity (PA) can affect health negatively.

Pg. 1, Ln. 17: Rephrase to “To achieve prevent detrimental health outcomes, adolescents must be physically active and aim to meet PA guidelines.”

Pg. 1, Ln. 18: What makes AC to school “easiest”?

 Pg. 1, Ln. 20-21: Rephrase to explicitly state study objectives and/or research question. It would be important to clearly state independent and dependent variables that will be tested. There is also a lack of presentation regarding the analysis conducted. Please also state how data was analyzed.

Introduction

Pg. 1, Ln. 40-41: Again, I would be cautious with the word “easiest”

I am concerned AC to schools is being framed in this study as a “solution” to physical inactivity. For instance, can you share substantial evidence of the predictive power of AC to schools on MVPA?

On perceived/social norms: Do the authors believe that perceived/social norms can be a target concept in this age group that could turn beneficial in promoting greater AC to schools? By how much? And do the authors believe greater promotion of AC to schools will significantly impact adolescents’ physical inactivity to prevent disease?

Methods:

Pg. 2, Ln. 79: “Questionnaire”? What questionnaire? First, how was the study sample recruited (not just from where)? Were there any screening procedures? Where does the questionnaire comes from? Has it been validated? Did the study receive IRB approval?

Pg. 2, Ln. 83: “2.2 Assessments and Measures” Please indicate which is your dependent variable and independent variable (s). Provide significant details on which indicators from your questionnaire specifically measured your dependent and independent variable (s). Add details on any psychometric work done prior to main analysis.

Results:

This section is difficult to review. I can understand that the authors were able to assess a degree of “prevalence” related to Czech adolescents’ AC to schools. Authors also chose to “correlate” two indicators (one “objective” and another subjective) regarding AC to schools. They are “weakly” associated. Then what comes to mind for me is the typical “so what?” question. I do not think that the proposed science provided sufficient evidence about how perceived norms influence AC to schools among Czech adolescents.

Discussion:

Pg. 5, Ln. 139-140: I can appreciate the power of this statement. I definitely agree that information from your questionnaire could be disseminated to schools as reports and such. Potentially, it could serve as a baseline assessment that leads to additional efforts.

Pg. 5, Ln. 147-148: “Those who do not commute actively tend to estimate their peers to be inactive rather than active commuters.” I don’t agree with this statement. It could also be the other way around. Those who have perceptions about peers’ inactivity could also tend to not commute. I would re-phrase.

Pg. 5, Ln. 164: Remove “that”

“Social norms theory” Please re-visit what you mean by this. I would recommend that the authors actually adopt a theoretical lens that could be tested with the appropriate dataset and statistical approach.

Author Response

Dear reviewer, thank you for your valuable comments. They have been of great help to us during the revision of the manuscript.

As correction of English has been suggested, the manuscript was reviewed by a native speaker from the Editage.com.

For better orientation, we are providing a point-by-point responses to your comments and suggestions.

Point 1: Title: “The Role of Perceived Norms in Adolescents Actively Commuting to Schools” What is your outcome? It’s not clear at first glance from reading your title.

Response 1: Thank you for your note. Our former title could have been misleading. Therefore, we have amended it as follows: How do Czech Adolescents Perceive Active Commuting to School: A Cross-Sectional Study.

Point 2: Abstract: Rephrase the first statement to: A lack of daily physical activity (PA) can affect health negatively.

Response 2: Thank you for your comment. The Abstract as a whole has been  restructured and rephrased substantially, including the first statement.

Point 3: Pg. 1, Ln. 17: Rephrase to “To achieve prevent detrimental health outcomes, adolescents must be physically active and aim to meet PA guidelines.”

Response 3: Thank you, we have rephrased Pg. 1, Ln. 17. and the first statement of abstract according to your suggestions.

Point 4: Pg. 1, Ln. 18: What makes AC to school “easiest”?

Response 4: You are right, that it was a too strong statement. We have replaced it by: “One of the most natural ways…”

Point 5: Pg. 1, Ln. 20-21: Rephrase to explicitly state study objectives and/or research question. It would be important to clearly state independent and dependent variables that will be tested. There is also a lack of presentation regarding the analysis conducted. Please also state how data was analyzed.

Response 5: Thank you for your comment. As indicated above, the Abstract has been completely rewritten .

Point 6: Pg. 1, Ln. 40-41: Again, I would be cautious with the word “easiest”

Response 6: You are right, that it was too strong statement. We have replaced it by: “One of the most natural way…”

Point 7: I am concerned AC to schools is being framed in this study as a “solution” to physical inactivity. For instance, can you share substantial evidence of the predictive power of AC to schools on MVPA?

Response 7: We acknowledge that the mentioned statement was too strong. For this reason, we have reworded it: “One of the ways to contribute to obtaining this amount of PA is…” and we have also added one more recent study (Salway et al., 2019) which concluded that AC was associated with greater physical activity on the day. There are many other studies that demonstrate positive correlation between AC and PA/compliance with PA recommendations (Kohl, H.W.; Craig, C.L.; Lambert, E.V.; Inoue, S.; Alkandari, J.R.; Leetongin, G.; Kahlmeier, S. The pandemic of physical inactivity: Global action for public health. Lancet 2012, 380, 294–305., Slingerland, M.; Borghouts, L.B.; Hesselink, M.K.C. Physical activity energy expenditure in Dutch adolescents: Contribution of active transport to school, physical education, and leisure time activities. J. Sch. Health 2012, 82, 225–232.,., etc.) but we wanted to avoid excessive number of citations. We have also relocated the sentence: “Previous findings confirmed that AC to schools contributed to the successful achievement of health recommendations for PA in Czech adolescents [15]” to support this idea.

Point 8: On perceived/social norms: Do the authors believe that perceived/social norms can be a target concept in this age group that could turn beneficial in promoting greater AC to schools? By how much? And do the authors believe greater promotion of AC to schools will significantly impact adolescents’ physical inactivity to prevent disease?

Response 8: Thank you for the useful remarks. Indeed, we are curious whether the social norms approach could be useful in promoting AC. Answering that question is one of the aims of our study. To make it clearer to readers, we have added the following paragraphs in the Introduction:

“Human behavior is profoundly affectedby social norms influences on across the lifespan (Rice and Klein 2019). According several theories of health behavior, social comparisons and perceived social norms affect behaviour in a wide variety of domains (e.g., Schulz et al 2007; Miller and Prentice 2016; Klein and Rice 2020). According to meta-analysis of Sheeran et al (2016) interventions that modify attitudes, norms, and self-efficacy are effective in promoting health behavior change. They found that experimentally induced changes in attitudes, norms, and self-efficacy all led to medium-sized changes in intention (d+ = .48, .49, and .51, respectively), and engendered small to medium-sized changes in behavior (attitudes-d+ = .38, norms-d+ = .36, self-efficacy-d+ = .47).

Recent reviews suggest that peer norms have the potential to shape PA during adolescence in particular (Salvy et al 2012.; Sawka et al 2013; Draper et al. 2015). The effect of social norms on adolescent health-related behaviors was best described in risky behaviors, such as alcohol consumption, tobacco use and substance use (Stock 2016, Cooke, Dahdah, Norman, & French, 2016, Dempsey et al 2016, Helmer 2014, Pischke 2012). Nevertheless some previous studies indicate that approaches based on social norms could be a suitable method of intervention to combat physical inactivity or promote PA.  (Baker, Little, & Brownell, 2003, Kim, J. et al. 2017, Draper 2015).

The Social Norms Approach (SNA) is the most widely used intervention strategy for promoting positive health-related behaviors based on social norms effect (Dempsey 2018). The approach is founded on the premise that individuals misperceive their peers’ behaviors and attitudes, with evidence of under- and over-estimations of behaviors and peer approval for a range of positive and negative behaviors, respectively (Helmer 2016). The greater these misperceptions are, the more likely an individual is to engage in negative behaviors, such as consuming heavier amounts of alcohol and reduce positive behaviors, such as eating healthily, using sun protection or being physically active.”

Point 9: Pg. 2, Ln. 79: “Questionnaire”? What questionnaire? First, how was the study sample recruited (not just from where)? Were there any screening procedures? Where does the questionnaire comes from? Has it been validated? Did the study receive IRB approval?

Response 9: Thank you for all these points. Information about sample recruitment has been revised and rewritten in more detail (including main inclusion/exclusion criteria):

“The present cross-sectional study involved 1586 adolescents (52% of boys) aged 11–15 years (mean 12.95 ± 3.6 years) from 12 randomly selected elementary schools (Grades 6–9) from the Czech Republic. Schools with a specific focus on sport and schools for pupils with special educational needs were not recruited. The main inclusion criteria for the recruitment of participants were defined by age and good health condition. Those participants who reported medical complications that could affect their AC behavior were excluded from the study.”

The description of the questionnaire and general information about its validation have been incorporated:

“The questionnaire on activities was developed for the purpose of the SONIAA (Social norms intervention in the prevention of excessive sitting and physical activity promotion among Czech adolescents) project and was compiled from various internationally recognized surveys (e.g., HBSC, IPEN Adolescent, and YAP). The questionnaire was designed to measure various domains of PA and sedentary behavior with a focus on respondents’ own behaviors and attitudes, as well as perception of PA and sedentary behavior in their peers..For the purpose of this study, we used question on AC to school over the past seven days, which was adapted from the YAP (Youth Activity Profile) as a feasible and valid survey instrument at the group-level [27]: “How many days did you walk, bike or ride scooter/skateboard to school?”. The students could select a response from five options, ranging from “zero days (never) to four or five days (almost every day)”.

Perceptions of peers’ AC was assessed using the following item “How many days do you think did most of your classmates walk, bike, or ride a scooter/skateboard to school?”. Moreover, we collected data on perceived peer attitudes towards peer AC using question “Do you think is it fine like that? We are asking you about your opinion on active time spent on the way to school by majority of your classmates.” Response options were ‘Yes, it is fine, ‘Rather fine’, ‘I don't know’, ‘Rather not fine’, and ‘No, it is not fine’. “

In addition, anew chapter 2.3 Procedure has been created:

“Data were collected from 2017 to 2018 in regular school weeks during the spring and autumn seasons. The pupils filled in an electronic questionnaire at school during the class under the supervision of teachers and researchers.”

Finally, information about ethics committee approval has been added in the manuscript in the chapter 2.1 Procedure Sample and Participant Selection:

“The study was approved under reference number 38/17 by the Ethics Committee of the Faculty of Physical Culture, Palacký University Olomouc, which is governed by the ethical standards set out in the World Medical Association Declaration of Helsinki and its later amendments.”

Point 10: Pg. 2, Ln. 83: “2.2 Assessments and Measures” Please indicate which is your dependent variable and independent variable (s). Provide significant details on which indicators from your questionnaire specifically measured your dependent and independent variable (s). Add details on any psychometric work done prior to main analysis.

Response 10: Our main finding (according to the main aim) is in a form of description of perceived AC and differences from the actual AC. In the second supportive analysis (correlation), own frequency of AC acted as an independent variable, while dependent variable was represented by perceived frequency of their classmates’ AC. This information has been also explicitly stated in the 2.4 section on data processing.

Point 11: This section is difficult to review. I can understand that the authors were able to assess a degree of “prevalence” related to Czech adolescents’ AC to schools. Authors also chose to “correlate” two indicators (one “objective” and another subjective) regarding AC to schools. They are “weakly” associated. Then what comes to mind for me is the typical “so what?” question. I do not think that the proposed science provided sufficient evidence about how perceived norms influence AC to schools among Czech adolescents.

Response 11: Thank you for this point. According to your comments and suggestions, we rethinked a message of this paper a bit (including a change of the title). Our main message is no more standing on only that “weak” association (coming from an “objective” and another subjective indicators) and the analyses where we were not able to identify the direction of the relationship. Our main message is coming from a significant difference (for girls, boys and all grades) between self-reported and perceived AC in classmates. There is also an important fact, that AC is reported by more than 50% of students, but they are convinced that it is less than 50% among their classmates. According to the social norms theory – what is over 50% (i.e. majority), that becomes a social norm. Because of this difference, there is a potential that appropriately focused informative feedback could be effective in the way of increasing AC / PA.     

Point 12: Pg. 5, Ln. 139-140: I can appreciate the power of this statement. I definitely agree that information from your questionnaire could be disseminated to schools as reports and such. Potentially, it could serve as a baseline assessment that leads to additional efforts.

Response 12: Thank you for this remark. We agree, it is the starting point for our two ongoing projects.  Indeed, it is planned that survey presented in the manuscript should serve as a baseline measurement. Next steps include, providing schools and pupils, in particular, with the feedback and carry out a follow-up measurement.

Point 13: Pg. 5, Ln. 147-148: “Those who do not commute actively tend to estimate their peers to be inactive rather than active commuters.” I don’t agree with this statement. It could also be the other way around. Those who have perceptions about peers’ inactivity could also tend to not commute. I would re-phrase.

Response 13: You are right that from our data and analyses we are not able to draw a conclusion on the direction  of the relationship. We have changed “our finding also indicate” to “may indicate”. On the other hand, we believe that basic principle “judging others by one’s own standards” works in this situation. We have reworded the text and added “vice versa” and also mentioned it as a limitation: “Moreover, we were not able to analyze the direction of the relationship between own actual level of AC and perceived AC because of the cross-sectional nature of our data.”

Point 14: Pg. 5, Ln. 164: Remove “that”

Response 14: Thank you, it was removed.

Point 15: “Social norms theory” Please re-visit what you mean by this. I would recommend that the authors actually adopt a theoretical lens that could be tested with the appropriate dataset and statistical approach.

Response 15: Manuscript, especially introduction, has undergone a substantial amendment. New paragraphs about social norms have been added. We hope that now the information is clearer .

Reviewer 2 Report

The first part of this manuscript is well written and easy to follow. The research question is interesting, although one of the main problem with the data is the lack of control for travelled distance, as also stated in the main limitations. 

As for the Results section, I suggest adding the figure with the correlation. 

As for the Discussion, this section needs to be expanded and English language revised. For example, the Authors did not discuss the gender difference in their results. Appropriate comparison with the literature should also be made.

Author Response

Dear reviewer, thank you for your valuable comments. They have been of great help to us during the revision of the manuscript.

As correction of English has been suggested, the manuscript was reviewed by a native speaker from the Editage.com.

For better orientation, we are providing a point-by-point responses to your comments and suggestions.

 Point 1:The first part of this manuscript is well written and easy to follow. The research question is interesting, although one of the main problem with the data is the lack of control for travelled distance, as also stated in the main limitations. 

Response 1: We are aware of a shortcoming in the data when we did not determine the exact distance for active transport and that is why we had mentioned it in the paragraph presenting the limitations. However, any actively covered distance is always more positive than a passive alternative e.g. using a car or public transport. Previous studies reported that distance to school affects active/passive transport.

Point 2:As for the Results section, I suggest adding the figure with the correlation. 

Response 2: We appreciate your suggestion. We discussed this option with our statisticians and concluded that this correlation is not the main result of this study but only a supportive analysis. For this reason, we prefer to communicate only main ideas with figure 1 and figure 2. In addition, figures with the correlation are more common for continuous variables.

Point 3: As for the Discussion, this section needs to be expanded and English language revised. For example, the Authors did not discuss the gender difference in their results. Appropriate comparison with the literature should also be made.

Response 3: We have added a comparison of our results according to the gender differences and also comparison of active commuting with former studies.

Reviewer 3 Report

In the abstract section:

This reviewer recommends to change this sentence “I will apply the social norms theory” into a passive sentence.

The aim of this study is not only “to investigate how adolescents perceive AC levels among their classmates” but also about themselves AC levels.

In keywords section:

So as to increase the visibility of this paper, it is important not to repeat keywords that already are in the title.

In introduction section:

Some light language mistakes are shown in the whole paper. Please try to review it.

In the line 64 there is an opening “ without and ending “.

Author/s should consider the possibility of explaining what levels of PA are recommended in order to be considered physically active. The World Health Organisation could be an appropriate institution with this regard.

In 2.1. section.

Was this study approved by any ethical committee?

Author/s should indicate the design of this research (transversal, descriptive…) based on a research manual.

In results section:

Author/s may include an analysis about how the differences change regarding the grade (if this change is significant or not, perhaps only among boys or only among girls).

In discussion section:

This is the weakest part of the paper. There are only two quotes so the real discussion between the achieved results and the already published literature is, by far, not enough. In this section it is important not only to indicate if the results coincide or not with other research but also to try to explain their causes and their pedagogical implications. This section must be deeply improved.

In conclusion section:

Author/s should explicitly indicate the aim and what they can conclude about it.

In references section:

Author/s should try to increase the amount of studies from 2020 and 2019 or, at least, from the last five years. It is important to update the references.

Author Response

Dear reviewer, thank you for your valuable comments. They have been of great help to us during the revision of the manuscript.

As correction of English has been suggested, the manuscript was reviewed by a native speaker from the Editage.com.

For better orientation, we are providing a point-by-point responses to your comments and suggestions.

 Point 1:This reviewer recommends to change this sentence “I will apply the social norms theory” into a passive sentence.

Response 1: Thank you, it has been changed.

Point 2:The aim of this study is not only “to investigate how adolescents perceive AC levels among their classmates” but also about themselves AC levels.

Response 2: Thank you for your comment. We have reworded  the aim as follows: “ Given that discrepancy between actual and perceived levels of behaviour are crucial to application of the SNA, our primary aim  will be was to investigate whether Czech adolescents misperceive their peers’ AC behaviors and attitudes towards  AC. Furthermore, we will aimed to investigate the the associations between  AC perception and their actual  AC behaviors and whether these associations differed by gender and class grade..” Also, we have incorporated basic information about analysis conducted in the abstract: “ Basic descriptive statistics, the chi-square tests and correlation analysis were used to analyze the data..” Detailed information about independent and dependent variables and analysis conducted is written in the data processing section.

Point 3:So as to increase the visibility of this paper, it is important not to repeat keywords that already are in the title.

Response 3: You are right. We have selected a  different set of keywords: school-aged children, cycling, walking, active transport, misperceptions, social norms

Point 4:Some light language mistakes are shown in the whole paper. Please try to review it.

Response 4: As you suggested English changes, the manuscript was reviewed by a native speaker from the Editage.com.

Point 5:In the line 64 there is an opening “ without and ending “.

Response 5: Thank you for this point. It was corrected: “correction of misperceptions”

Point 6:Author/s should consider the possibility of explaining what levels of PA are recommended in order to be considered physically active. The World Health Organisation could be an appropriate institution with this regard.

Response 6: We agree that such an information is essential. For this reason, it had been included in the very first paragraph of the Introduction.

Point 7: Was this study approved by any ethical committee?

Response 7: Yes, thank you for this point. Information has been added in the manuscript in the section 2.1 Procedure Sample and Participant Selection: “The study was approved under reference number 38/17 by the Ethics Committee of the Faculty of Physical Culture, Palacký University Olomouc, which is governed by the ethical standards set out in the World Medical Association Declaration of Helsinki and its later amendments.”

Point 8: Author/s should indicate the design of this research (transversal, descriptive…) based on a research manual.

Response 8: Thank you for this comment. The cross-sectional nature of the study design is now reflected in the title: “How do Czech Adolescents Perceive Active Commuting to School: A Cross-Sectional Study.” Furthermore, it has been added in the first paragraph of the Materials and Methods section: “The present cross-sectional study involved... “

Point 9:Author/s may include an analysis about how the differences change regarding the grade (if this change is significant or not, perhaps only among boys or only among girls).

Response 9: We had already discussed this option during initial data analyses but we would like to keep separate analyses –  one for gender and one for grades. For clear description, we have changed the sentence to: “the difference between actual level of AC and perceived level of their peers‘ AC was statistically significant (p>0.01) across all grades regardless of gender.”

Nonetheless, we have conducted suggested analysis and inserted the following text to The Results section: “The average difference between the students’ perceived and actual norm of AC is 25% (64% vs. 39%) for boys ( (χ2 = 103.28; p < .001) and 23% (72% vs. 49%) for girls (χ2 = 84.06; p < .001). There is a smaller difference across younger (6th + 7th grade) students (62% vs. 43%; χ2 = 60.69; p < .001) than older (8th + 9th grade) students (70% vs. 44%; χ2 = 102.94; p < .001).”

We appreciate your comment because the difference between younger and older students is a new and interesting fact for us. It appears that the adolescents become more active in terms of commuting, but the perception level is the same. Thank you for this inspirative suggestion.

We have also added the results of the chi-square analyses to the descriptive tables (Tables 1 and 2).

Point 10: This is the weakest part of the paper. There are only two quotes so the real discussion between the achieved results and the already published literature is, by far, not enough. In this section it is important not only to indicate if the results coincide or not with other research but also to try to explain their causes and their pedagogical implications. This section must be deeply improved.

Response 10: The Discussion section has undergone a very substantial reconstruction. Some parts of the text have been rephrased and we have also added two new paragraphs. Conclusions have been completely rewritten too.

Point 11: Author/s should explicitly indicate the aim and what they can conclude about it.

Response 11: We have rewritten the Conclusions section according to your comment: “Our primary aim was to investigate whether Czech adolescents misperceived their peers’ AC behaviors and attitudes towards AC to assess the potential of intervention based on SNA. Our study points out that there is a discrepancy between actual level of AC and perceived level of peers’ AC. Adolescents overestimate the prevalence of passive transport despite that most of them commute actively. These findings indicate that there might be a room for targeted intervention based on SNA to increase the PA of adolescents or at least strengthen their actual positive behavior..”

Point 12: Author/s should try to increase the amount of studies from 2020 and 2019 or, at least, from the last five years. It is important to update the references.

Response 12: Thank you for raising this point. We have replaced some studies in the manuscript with newer publications.

Round 2

Reviewer 3 Report

Dear author/s,

this reviewer wants to congratulate you since the paper has been significantly improved.

Kind regards.

Author Response

Dear reviewer,

thank you very much for your previous comments and suggestions.

We are very thankful that you helped us to improve that article.

Best Regards,

Team of Authors